# The *aroA* and *luxS* Double-Gene Mutant Strain Has Potential to Be a Live Attenuated Vaccine against *Salmonella* Typhimurium

**DOI:** 10.3390/vaccines12020162

**Published:** 2024-02-04

**Authors:** Wei Zuo, Denghui Yang, Xiaojun Wu, Beibei Zhang, Xinyu Wang, Jiangang Hu, Jingjing Qi, Mingxing Tian, Yanqing Bao, Shaohui Wang

**Affiliations:** Shanghai Veterinary Research Institute, Chinese Academy of Agricultural Sciences, Shanghai 200241, China; zuo15910132132@sina.com (W.Z.); yaonan2345@sina.com (D.Y.); xjwu1993@sina.com (X.W.); bzhang1997@sina.com (B.Z.); xwang0630@sina.com (X.W.); hjg@shvri.ac.cn (J.H.); qijingjing@shvri.ac.cn (J.Q.); mxtian@shvri.ac.cn (M.T.)

**Keywords:** *Salmonella* Typhimurium, double mutation strains, live attenuated vaccine

## Abstract

*Salmonella* Typhimurium (*S.* Typhimurium) is a zoonotic pathogen posing a threat to animal husbandry and public health. Due to the emergence of antibiotic-resistant strains, alternative prevention and control strategies are needed. Live attenuated vaccines are an ideal option that provide protection against an *S.* Typhimurium pandemic. To develop a safe and effective vaccine, double-gene mutations are recommended to attenuate virulence. In this study, we chose *aroA* and *luxS* genes, whose deletion significantly attenuates *S.* Typhimurium’s virulence and enhances immunogenicity, to construct the double-gene mutant vaccine strain SAT52ΔaroAΔluxS. The results show that the mutant strain’s growth rate, adherence and invasion of susceptible cells are comparable to a wild-type strain, but the intracellular survival, virulence and host persistence are significantly attenuated. Immunization assay showed that 10^6^ colony-forming units (CFUs) of SAT52ΔaroAΔluxS conferred 100% protection against wild-type challenges; the bacteria persistence in liver and spleen were significantly reduced, and no obvious pathological lesions were observed. Therefore, the double-gene mutant strain SAT52ΔaroAΔluxS exhibits potential as a live attenuated vaccine candidate against *S.* Typhimurium infection.

## 1. Introduction

*Salmonella* Typhimurium (*S.* Typhimurium) is one of the most common zoonotic pathogens around the world. Domestic animals are a major reservoir of *S.* Typhimurium; the infection commonly comes with mild symptoms [1,2]. Stress factors like high stocking rates, prolonged nutrition deficiency, immunity suppression and chronic pathogen infection can induce outbreaks of salmonellosis, leading to diarrhea, abortion, respiratory disease or even fatal septicemia, which significantly reduce animal productive performance [3]. Humans are usually infected with *S.* Typhimurium by consuming contaminated animal products like eggs, meat and milk. The infection causes salmonellosis with symptoms like vomiting, diarrhea, abdominal pain and fever, leading to high mortality rates in developing and undeveloped countries [4]. Thus, *S.* Typhimurium poses a great threat to animal husbandry, food safety and public health [5,6].

Currently, antibiotics are widely used in animal husbandry to control and treat *S.* Typhimurium infections, but the prevalence of drug- and multi-drug-resistant strains urges alternative strategies [7,8]. In related immunology studies, antibodies and primed T-cells confer protection against this challenge, indicating a vaccine’s potential in controlling animal *S.* Typhimurium infections [9,10,11]. Also, vaccinated animals shed a significantly lower bacterial burden in eggs, meat and offal, leading to a reduction in human *S.* Typhimurium infection rates [12]. Thus, vaccination is important to food safety and public health by interrupting *S.* Typhimurium transmission from animals to humans [13]. Currently, inactivated strains, live attenuated strains and Vi polysaccharide have been developed as *Salmonella* vaccines. The live attenuated strains are favored due to their stimulation of humoral and cellular immune responses, which are necessary for sustainable immunity against intracellular pathogens [14]. In earlier studies, the Ty21a strain was produced with chemical mutagenesis. With deficiency in the *galE* gene, it exhibits a rough O-PS phenotype and significantly attenuated virulence to hosts [15]. Later, with the development of site-directed mutant methods, as well as knowledge of *Salmonella* virulence genes, researchers were able to generate strains with accurate mutations at the desired gene locus, which facilitates the production of genetically engineered live attenuated strains [16,17].

Mostly, genes related to *S.* Typhimurium’s metabolism, secretion systems, regulators and protein degradation are selected as vaccine candidates. This includes *aroCD* in the aromatic acid biosynthesis pathway [18], *ssaV*, which encodes the needle apparatus for T3SS [19], two-component regulatory systems encoding genes *phoP*/*phoQ* [20] and *clpPX*, which regulates flagellin expression by degrading FlhD/FlhC [21]. Though single mutations of those genes reduces *S.* Typhimurium’s virulence to the host and provides effective protection against wild-type strain challenges, administration requires two independent mutations to avoid the risk of virulence reversion [14]. Several double-gene mutation strains’ potential as live attenuated vaccines were evaluated. With the deletion of *aroC* and *aroD*, CVD 908′s virulence in animal models was further attenuated and successfully induced humoral immune response [22]. In *S.* Typhimurium wild-type strain SR-11, the deletion of *sucCD* and *frdABCD* exhibits even lower virulence to hosts than single mutant strains, and peroral inoculation confers protection against the challenge of the parental strain [23]. This indicates that double-gene mutations yield safer live attenuated vaccines with guaranteed protection efficacy.

In our previous study, the deletion of both *aroA* and *luxS* would further attenuate *Escherichia coli* virulence in susceptible hosts without a significant impact on its immunogenicity [24]. It intrigued us to explore these two genes’ potential in developing a live attenuated vaccine against *S.* Typhimurium. In this study, we introduced *aroA* and *luxS* double-gene mutation in *S.* Typhimurium strain SAT52 with λ-Red-mediated recombination [25]. SAT52ΔaroAΔluxS exhibited significantly attenuated persistence in mice and induced strong humoral and cellular immune responses. Intraperitoneal inoculation conferred protection against SAT52’s challenge, demonstrating the potential of SAT52ΔaroAΔluxS as a live attenuated vaccine candidate.

## 2. Materials and Methods

### 2.1. Ethics Statement

Mice were housed under standard conditions of temperature and relative humidity with free access to food and water without any antibiotics. A 12 h light/dark cycle was applied throughout the study to simulate natural diurnal conditions. The animal experiments were approved by the Animal Care and Use Committee at the Shanghai Veterinary Research Institute, Chinese Academy of Agricultural Sciences (CAAS). All animals were handled in accordance with the guidelines of the Institutional Administrative Committee (SYXK[Hu]-2020-0027).

### 2.2. Bacterial Strains, Primers, Plasmids and Cell Lines

The primers used in this study are listed in Table 1. The *S.* Typhimurium wild-type strain SAT52 was used for attenuated vaccine construction [26]. All the *S.* Typhimurium strains were cultured in Luria-Bertani (LB, Sigma-Aldrich, St Louis, MO, USA) media at 37 °C with aeration. When necessary, ampicillin (100 μg/mL, Sangon Biotech, Shanghai, China) or chloramphenicol (30 μg/mL, Sangon Biotech, Shanghai, China) was added to the media. The plasmids pKD46, pKD3 and pCP20 were preserved in our lab. They were used to knockout desired genes in *S.* Typhimurium with λ-Red homologous recombination method [25]. Murine macrophage RAW264.7 and human cervix epithelial cell HeLa were purchased from American type culture collection (ATCC, Manassas, VA, USA) and preserved in our lab. Both of them were cultured in Dulbecco’s modified Eagle media (DMEM, Gibco, Thermo Fisher Scientific, Beijing, China) media with 10% fetal bovine serum (FBS, Gibco, Thermo Fisher Scientific, Australia) at 37 °C with 5% CO_2_, and no antibiotic was added for cell culture.

### 2.3. Construction of Mutant Strain SAT52ΔaroAΔluxS

The double-gene mutation strain was constructed according to the λ-Red recombinase method as described previously [26], with some modification and appropriate primers (Table 1). Briefly, first, the *aroA* and *luxS* genes were replaced with a chloramphenicol resistance gene cassette. Then, the helper plasmid pCP20 was used to eliminate the cassette and achieve scarless deletion at desired genes. The mutant strains SAT52ΔaroAΔluxS were selected and confirmed by PCR and sequencing. 

### 2.4. Growth Assay

To compare the growth rates between SAT52ΔaroAΔluxS and SAT52, the growth kinetics of both strains were determined as described previously [27]. The freshly cultured bacteria was diluted to OD_600nm_ = 1.0. Then, 1 mL of the diluted bacteria culture was inoculated into 100 mL LB media, respectively. Both strains were cultured at 37 °C with shaking, and the OD_600nm_ was monitored at 1 h intervals for 16 h.

### 2.5. Bacterial Infection In Vitro

The in vitro infection assays were carried out as previously described [28]. For adherence and invasion, the HeLa cells were infected with SAT52 or SAT52ΔaroAΔluxS at MOI of 100, respectively. After 1 h of infection, the cells were washed twice with phosphate buffered saline (PBS, Gibco, Thermo Fisher Scientific, Beijing, China) to remove non-adherent bacteria. For adherence, the cells were lysed with 0.5% Triton X-100, the solution was used for 10-fold serial dilution and dispersed onto LB agar plates to enumerate the bacterial CFUs. For invasion, the infected cells were treated with 100 μg/mL gentamicin (Sangon Biotech, Shanghai, China) for another 1 h to kill extracellular bacteria, and then the cells were lysed as above to enumerate the bacterial CFUs.

For intracellular survival, the RAW264.7 cells were infected with SAT52 or SAT52ΔaroAΔluxS at MOI of 10, respectively. Similarly to invasion assay, the cells were infected for 1 h and treated with gentamicin to kill extracellular bacteria. After that, the cells were cultured in DMEM with 10 μg/mL gentamicin for 12 h. Then, the cells were washed twice with PBS to remove residual gentamicin and lysed as above to enumerate the CFUs of surviving bacteria.

### 2.6. LD50 Determination

To assess the virulence of double-gene mutation strain, the comparison of 50% median lethal dose (LD_50_) between SAT52 and SAT52ΔaroAΔluxS was conducted on 6-week-old BALB/c mice. For either strain, 48 mice were evenly divided into 6 groups (8 per group). Then, each mouse in every group was intraperitoneally injected with 1 × 10^4^, 1 × 10^5^, 1 × 10^6^, 1 × 10^7^, 1 × 10^8^ and 1 × 10^9^ CFUs of bacteria, respectively. The mice in negative control group were injected with PBS. The mortality rate was monitored until 14 days post-infection. The LD_50_ of each strain was calculated using the Spearman–Karber method [29]. 

### 2.7. In Vivo Persistence Assay

To assess bacteria’s persistence in mouse, two groups of 6-week-old BALB/c mice were immunized with 1 × 10^5^ and 1 × 10^6^ CFUs of SAT52ΔaroAΔluxS, respectively. The mice in negative control group were injected with PBS. The changes in body weight and disease symptoms were observed throughout the assay. At 2, 4, 6, 8, 10, 12 and 14 days post-immunization (dpi), 5 mice from each group were sacrificed, respectively. Their liver and spleen samples were aseptically collected, weighed, and homogenized in 0.5% Triton X-100 solution. The serial 10-fold dilutions of tissue homogenates were plated on XLT4 agar (Oxoid, Thermo Fisher Scientific, Hampshire, UK) to enumerate the CFUs.

### 2.8. Humoral Immune Responses

To assess humoral immune response, the *S.* Typhimurium specific IgG antibodies in serum were determined by indirect enzyme-linked immunosorbent assay (ELISA) as previously described [30]. Briefly, the blood was collected from infraorbital veins of 3 mice at 0, 2, 4 and 6 weeks post-immunization (wpi). The serum was isolated using a centrifuge and kept at −70 °C until testing. The microtiter plates (Corning, Kennebunk, ME, USA) were coated with inactivated and sonicated SAT52. The serum samples were diluted and added as primary antibody, respectively. Goat anti-mouse IgG conjugated with alkaline phosphatase (Thermo Fisher Scientific, Waltham, MA, USA) was used as secondary antibody. The optical density was measured at 450 nm with an automated microplate reader. 

### 2.9. Cell-Mediated Immune Responses by Splenocyte Proliferation

To evaluate cellular immune response, whole-cell lysate of SAT52 was prepared as follows: First, SAT52 was cultured in 100 mL of LB media until OD_600nm_ reached around 1.0. Then, the bacterial pellet was collected by centrifuge at 5000 rpm for 10 min. After that, the pellet was resuspended with 10 mL of RPMI-1640 media (Gibco, Thermo Fisher Scientific, Beijing, China), and lysed with sonication. The lysate was pushed through 0.22 μm sterilized filter, then diluted in cell culture media (RPMI-1640 media with 10% FBS) at 1:10.

Four mice from each group of in vivo persistence assay were sacrificed at 3 wpi to harvest splenocytes. The cells were seeded into 96-well plates (WHB, Shanghai, China) at 1 × 10^5^ cells/well, and cultured with 200 μL of media added with SAT52 whole-cell lysate. After 72 h, the cell proliferative rate was determined with an MTT-based assay as previously described [31]. The stimulation index was calculated according to MTT assay kit’s manufacturer instructions. In addition, the splenocytes were seeded into 6-well plates (WHB, Shanghai, China) at 1 × 10^6^ cells/well and stimulated with 2 mL of media added with SAT52 whole-cell lysate for 24 h. Then, total RNA was extracted from the stimulated cells using a RNeasy Plus mini kit (Qiagen, GmbH, Hilden, Germany) and subsequently converted into cDNA using a High Capacity cDNA Reverse Transcription Kit (Applied Biosystems, Foster, CA, USA). The relative mRNA level of IL-4 and IFN-γ were quantified with real-time RT-PCR as previously described, with mouse GAPDH as internal control gene [32].

### 2.10. Immune Protection Assay

The immune protective efficacy of SAT52ΔaroAΔluxS was evaluated in mouse. Six-week-old mice were randomly divided into 3 groups (16 per group) and were immunized with 1 × 10^5^ or 1 × 10^6^ CFUs of SAT52ΔaroAΔluxS, respectively. The mice in negative control group were injected with PBS. At 2 wpi, all mice were challenged with wild-type strain SAT52. The survival rate in each group was observed for 10 days after challenge. In addition, at 24 h post challenge, 5 mice of each group were sacrificed to determine the bacterial colonization in liver and spleen. Three mice’s livers and spleens were collected and fixed in 4% formaldehyde solution for histopathological examination as described in previous study [33]. 

### 2.11. Statistical Analysis

Statistical analysis were performed using the GraphPad Prism software 8.0.1 (San Diego, CA, USA). Data were presented as the mean ± SD. The differences among samples were analyzed with Student’s *t*-test, non-parametric Mann–Whitney *t*-test, and one-way or two-way ANOVA method, respectively. The *p* values less than 0.05 were considered as statistically significant.

## 3. Results

### 3.1. SAT52ΔaroAΔluxS Exhibited Reduced Bacterial Intracellular Survival in Cells

The *aroA* and *luxS* double-gene mutation was successfully constructed with *S.* Typhimurium SAT52 strain by λ-Red-mediated recombination. The coding regions of these two genes were replaced with FRT sequences, which disrupted their expression (Figure 1A). The final construction was free of antibiotic resistance gene cassette. The growth rate of the SAT52ΔaroAΔluxS and wild-type SAT52 were similar in LB liquid media within 16 h (Figure 1B), indicating that the deletion of *aroA* and *luxS* genes had no impact on the growth of SAT52ΔaroAΔluxS. Bacterial infection of host cells are essential for virulence. Thus, the adherence, invasion and intracellular survival capacities of SAT52ΔaroAΔluxS were investigated. On HeLa cells, although the adherence and invasion were slightly increased for SAT52ΔaroAΔluxS, the difference was not statistically significant when comparing with the wild-type strain (*p* > 0.05). However, SAT52ΔaroAΔluxS showed significantly attenuated survival in RAW264.7 cells (*p* < 0.001), which is a phenotype closely related to virulence (Figure 1C). 

### 3.2. Double Mutation of aroA and luxS Genes Attenuated SAT52 Virulence In Vivo

To assess the virulence of SAT52ΔaroAΔluxS, BALB/c mice were infected with different doses of bacteria to determine the LD_50_. The results showed that in SAT52 group, infection doses higher than 10^5^ CFUs lead to a 100% death rate of mice. An amount of 10^5^ CFUs caused a 62.5% death rate, and 10^4^ CFUs caused a 37.5% death rate. For SAT52ΔaroAΔluxS, only 10^8^ and 10^9^ CFUs caused an 87.5% and 100% death rate, respectively. No death was observed in lower infection doses. Thus, the LD_50_ of SAT52ΔaroAΔluxS and SAT52 was 4.22 × 10^7^ CFUs and 3.16 × 10^4^ CFUs, respectively. It clearly demonstrated the virulence of SAT52ΔaroAΔluxS was approximately 1335-fold lower than that of SAT52 (Table 2). 

### 3.3. SAT52ΔaroAΔluxS Could Be Cleaned by Hosts within Two Weeks

To determine whether attenuated virulence comes along with effective clearance by hosts, we intraperitoneally injected mice with 10^5^ and 10^6^ CFUs of SAT52ΔaroAΔluxS, respectively. Bacterial burden in liver and spleen were determined every other day within 2 weeks. The results showed that 10^6^ CFUs lead to higher bacterial burden at 2, 4, 6, 8 and 10 dpi in either liver or spleen. As time lapsed, the bacterial burden in two groups kept dropping with similar kinetics. At 12 dpi, SAT52ΔaroAΔluxS was not detected in mice liver and spleen (Figure 2A,B). No mice from PBS group were positive with *Salmonella*. In addition, no signs of body weight loss, ruffled fur, shivering or acute death were observed in mice throughout the assay, which indicated the attenuated virulence of SAT52ΔaroAΔluxS.

### 3.4. SAT52ΔaroAΔluxS Effectively Induced Immune Responses

To evaluate humoral responses induced by SAT52ΔaroAΔluxS, the level of IgG against *S.* Typhimurium whole-cell lysate in mice serum was determined with ELISA. The results showed that either 1 × 10^5^ or 1 × 10^6^ CFUs of SAT52ΔaroAΔluxS induced a significantly higher level of IgG antibody in serum as early as 2 wpi, and keeps increasing at 4 and 6 wpi. The 1 × 10^6^ CFUs induced higher IgG level than that of 1 × 10^5^ CFUs at all time points, which indicated that higher immunization dose yielded stronger humoral responses (Figure 3A). 

The cellular immune responses were evaluated by splenic lymphocyte proliferation assay and cytokines’ mRNA levels. As shown in Figure 3B, the stimulation index of both immunized groups were significantly higher than that of PBS group after stimulation with the SAT52 whole-cell lysate (*p* < 0.001). This indicated that SAT52ΔaroAΔluxS induced the development of memory lymphocytes, whose replication could be prompted upon corresponding antigens stimulation. In the SAT52ΔaroAΔluxS immunization group, the relative mRNA level of IFN-γ was upregulated by more than four-fold that of the PBS group. The relative mRNA level of IL-4 was upregulated around three-fold. Similarly to humoral responses, a higher immunization dose induced higher levels of cytokines’ mRNA (Figure 3C,D). This indicated that the immunization successfully established T-cell immunity responses. 

### 3.5. Vaccine Protection Efficacy

Mice were intraperitoneally injected with 10^5^ and 10^6^ CFUs of SAT52ΔaroAΔluxS, respectively. According to the in vivo persistence assay, the mice were challenged with 1 × 10^6^ CFUs of SAT52 at 2 wpi to avoid the influence of residual SAT52ΔaroAΔluxS. The survival curves showed that 1 × 10^6^ CFUs of SAT52ΔaroAΔluxS provided 100% protection throughout the 10-day experiment, while one mouse from the 1 × 10^5^ CFUs group was dead at 3 days post challenge, resulting in a 87.5% survival rate. In contrast, all the mice in the PBS group succumbed to SAT52’s challenge at 3 days post challenge (Figure 4A). The residual CFUs of SAT52 in immunized mice liver and spleen were around 100-fold lower than that of PBS group (Figure 4B,C), indicating that SAT52ΔaroAΔluxS effectively induced hosts immunity against SAT52 infection.

### 3.6. Histopathological Examination of Challenged Mice

Histopathological examination showed that no obvious pathological lesions were observed in the liver or spleen of immunized mice. However, in the liver of PBS group, filtration of inflammatory cells and formation of inflammatory foci were observed (indicated by arrows, Figure 5). The spleen of PBS group mice exhibited excessive reticular tissue proliferation and inflammatory cell filtration, disrupting the boundary of red and white pulps which could be clearly seen in the spleen of immunized mice (indicated by dashed lines, Figure 5). Taken together, SAT52ΔaroAΔluxS conferred protection against wild-type SAT52 and avoided the pathological damage caused by challenge.

## 4. Discussion

In previous studies, researchers have tried several methods to develop vaccines against *Salmonella* infection. Inactivated *Salmonella* was developed as a human vaccine, but it only provides protection against acute infection, and it induces symptoms like fever, headache and asthenia, which raises concerns about its safety [34,35]. Vi polysaccharide is a kind of widely applied human *Salmonella* subunit vaccines, and several commercially available Vi polysaccharide vaccines play important roles in preventing severe symptoms and containing *Salmonella* epidemic among undeveloped countries [36]. However, since *S.* Typhimurium has no capsule, the susceptible hosts could hardly benefit from Vi polysaccharide vaccination. In this way, live attenuated vaccines, which induce both humoral and cellular immune responses, became an ideal choice to defend *S.* Typhimurium infection, especially in animal husbandry [37,38,39]. In this study, with the λ-Red homologous recombination method, we successfully constructed a double-gene mutation strain SAT52ΔaroAΔluxS without residual antibiotic cassettes. The qRT-PCR showed that transcriptions of genes flanking *aroA* and *luxS* were not affected, indicating the specificity and reliability of our genetic deletion. SAT52ΔaroAΔluxS exhibited a similar growth rate, cell adherence and invasion capacity with parental strain SAT52, but its intracellular survival in RAW264.7 were significantly attenuated. In vivo experiment showed that SAT52ΔaroAΔluxS could be cleaned in 2 weeks without acute death in mice. Comparing with double-gene mutation strains from other studies, like SR11ΔfrdABCDΔsdhCDA, SAT52ΔaroAΔluxS caused no symptoms throughout the persistence assay, and provided better protection rate with much lower vaccination doses [23]. Immunization with SAT52ΔaroAΔluxS led to robust humoral and cellular responses. In the challenge assay, no pathological lesions were observed. For the group immunized with 1 × 10^6^ CFUs of SAT52ΔaroAΔluxS, no mice were killed by SAT52’s challenge. This clearly demonstrated potential of SAT52ΔaroAΔluxS as a live attenuated vaccine candidate. 

According to our best knowledge, no *S.* Typhimurium live attenuated vaccines have been developed with double-gene mutation on *aroA* and *luxS* genes. The *aroA* gene encodes a 3-phosphoshikimate 1-carboxyvinyltransferase to synthesize chorismate, an intermediate utilized by *Salmonella* to produce necessary nutrients like p-aminobenzoic acid (PABA) and 2,3-dihydroxybenzoic acid (DHB) for intracellular survival [40]. A lack of *aroA* leads to auxotrophic phenotype of *Salmonella* and significantly reduces the virulence, but maintains the immunogenicity, making *aroA* a common candidate gene for vaccine development [13,39]. Recently, a study in tumor therapy field demonstrated that a lack of *aroA* not only attenuated *Salmonella* metabolism, but also altered global gene expression profile and FljB phase 2 flagellin orientation on bacteria surface, which enhanced immunogenicity of the mutant strain [41]. This further stresses out the plausibility of knocking out *aroA* for vaccine development. However, the drawbacks to apply *aroA* mutants as live attenuated vaccines still remain, such as the induction of pathological changes and reduced body weights in tested avians [42,43]. To guarantee safety of vaccine strain, another gene mutation which further attenuates virulence should be included. *luxS* was chosen due to its deep involvement with *Salmonella* virulence phenotypes. By synthesizing 4,5-dihydroxy-2,3-pentanedione (DPD), LuxS stimulates quorum-sensing system AI-2’s activity and regulates multiple downstream target genes [44]. Among them, genes within *Salmonella* pathogenicity island 1 (SPI-1) are closely related to virulence phenotypes like invasion into susceptible cells and colonization in hosts. A lack of *luxS* significantly downregulates SPI-1 genes transcription, as well as *Salmonella* virulence [45]. As expected, the deletion of *luxS* raised SAT52ΔaroAΔluxS’s LD_50_ to 4.22 × 10^7^ CFUs, which is more than two-fold higher than that of the *aroA* single mutant strain (1.78 × 10^7^ CFUs) in our unpublished results, indicating a much attenuated virulence of the double-gene mutation strain. Also, a lack of *luxS* should have no negative impact on *Salmonella* immunogenicity [46]. Similarly to *aroA*, it affects flagellin by driving flagellar variation towards a more immunogenic phase 1, which might also enhance the mutant strain’s immunogenicity [47]. Comparing with *aroA* single mutant strains, which requires more than 1 × 10^7^ CFUs to achieve 100% protection rate in other studies [48,49], SAT52ΔaroAΔluxS only needed an immunization dose of 1 × 10^6^ CFUs per mouse and exhibited a higher protection efficacy. Taken together, as a vaccine candidate, the *aroA* and *luxS* double-gene mutation strain could guarantee immunogenicity with enough biological safety.

However, the application of live attenuated vaccines is not totally free of risks. In immunity-suppressed animals, even double-gene mutation strains could induce lethal infection [50]. If animal herds have been infected with immunity-suppressing pathogens, the use of live attenuated *S.* Typhimurium vaccines would lead to disease outbreaks instead of protection. Also, the abuse of antibiotics might eliminate live attenuated strains before establishment of immunity, therefore nullifying the protection offered by vaccination. This indicates that a successful immunization strategy cannot rely solely on vaccination, and close health surveillance and comprehensive feeding plans are also needed.

## 5. Conclusions

In conclusion, the live attenuated *S.* Typhimurium strain SAT52ΔaroAΔluxS was able to induce both strong humoral and cellular immune responses in mice. It also conferred effective protection against SAT52’s challenge. This study indicated SAT52ΔaroAΔluxS’s potential role as a live attenuated vaccine candidate against *S.* Typhimurium infection.

## Figures and Tables

**Figure 1 vaccines-12-00162-f001:**
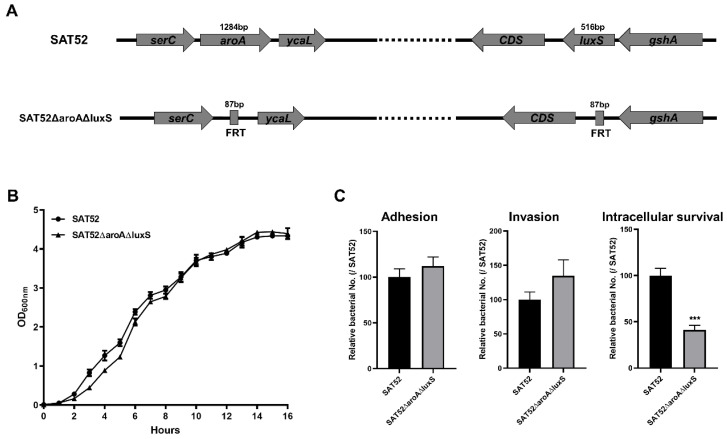
Characterization of double-gene mutation strain SAT52ΔaroAΔluxS. (**A**) Schematic of *aroA* and *luxS* gene mutation. The coding region of *aroA* and *luxS* genes were replaced with FRT sequences, respectively. (**B**) Growth rate of SAT52 and SAT52ΔaroAΔluxS in LB media. (**C**) Adherence and invasion in HeLa cells, and intracellular survival in RAW264.7 cells. Data are presented as mean ± SD (*n* = 3), and analyzed using a Student’s *t* test. *** *p* < 0.001.

**Figure 2 vaccines-12-00162-f002:**
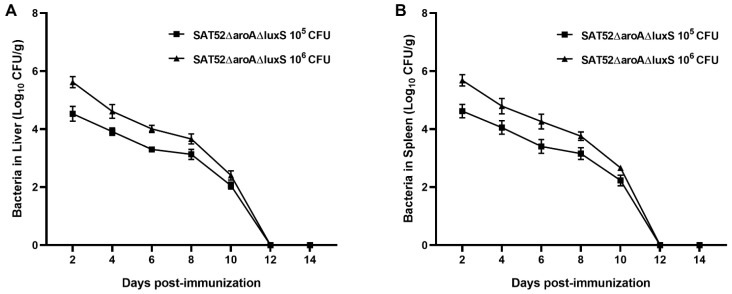
In vivo persistence assay of SAT52ΔaroAΔluxS. (**A**) Bacterial burden was determined in mice liver every other day. (**B**) Bacterial burden was determined in mice spleen every other day. SAT52ΔaroAΔluxS caused no symptoms or death and could not be detected at 12 dpi. Data are presented as mean ± SD (*n* = 5).

**Figure 3 vaccines-12-00162-f003:**
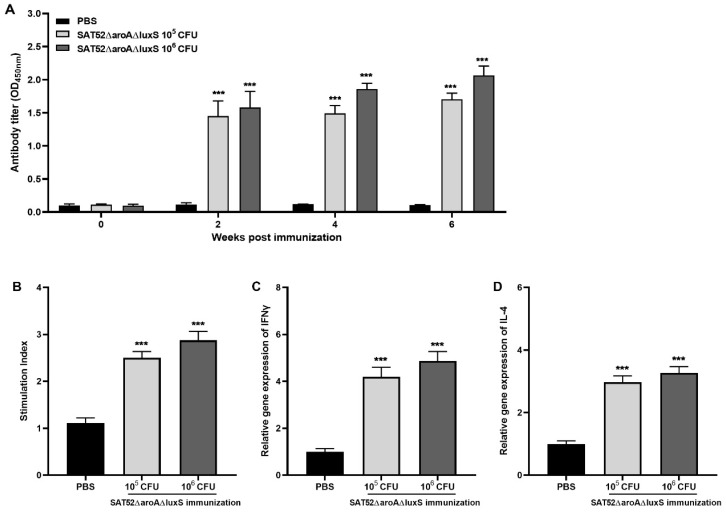
Immunogenicity of SATΔaroAΔluxS. (**A**) SATΔaroAΔluxS effectively induced IgG antibody against SAT52 at all time points. (**B**) Upon stimulation of SAT52 whole-cell lysate, splenocytes from SATΔaroAΔluxS-immunized groups exhibited significantly higher stimulation index when comparing with PBS group. (**C**) The relative mRNA levels of IFN-γ in splenocytes from SATΔaroAΔluxS-immunized groups were significantly higher than that of PBS group. (**D**) The relative mRNA levels of IL-4 in splenocytes from SATΔaroAΔluxS-immunized groups were significantly higher than that of PBS group. For IgG antibody assay in (**A**), the data are presented as mean ± SD (*n* = 3), and analyzed with two-way ANOVA, followed by Holm–Sidak multiple comparisons. For stimulation index, IFN-γ and IL-4 mRNA level in (**B**–**D**), the data are presented as mean ± SD (*n* = 4), and analyzed with one-way ANOVA, followed by Holm–Sidak multiple comparisons, respectively. *** *p* < 0.001.

**Figure 4 vaccines-12-00162-f004:**
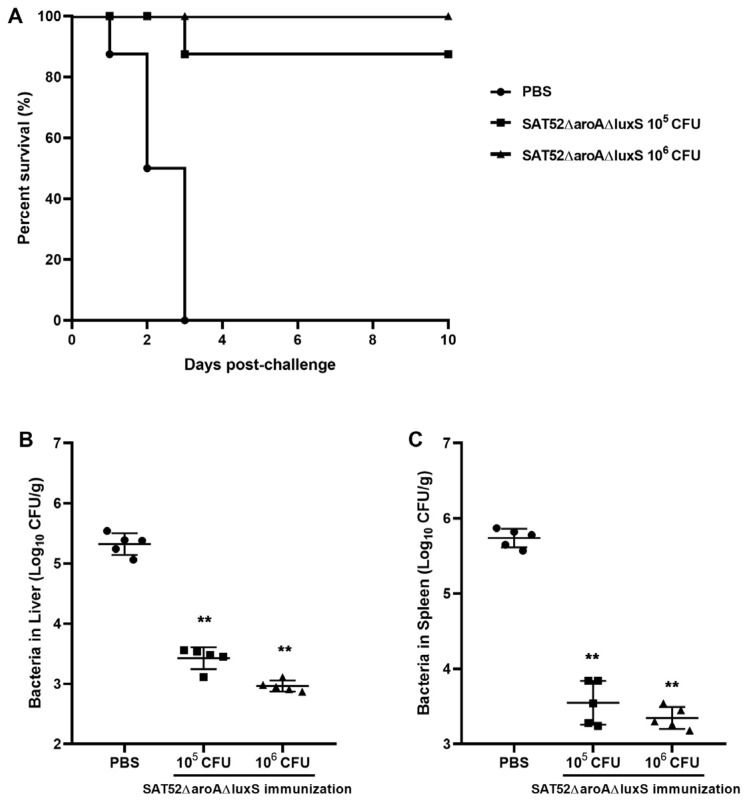
Evaluation of protection efficacy of SATΔaroAΔluxS. (**A**) Survival curves of mice after SAT52’s challenge, which demonstrated that 10^6^ CFUs of SATΔ*aroA*Δ*luxS* provided better protection against SAT52. (**B**) Bacterial burden in liver at 24 h post challenge. (**C**) Bacterial burden in spleen at 24 h post challenge. The data in (**B**,**C**) were presented as mean ± SD (*n* = 5). The significance between immunization groups and PBS group were analyzed with non-parametric Mann–Whitney *t*-test, respectively. ** *p* < 0.01.

**Figure 5 vaccines-12-00162-f005:**
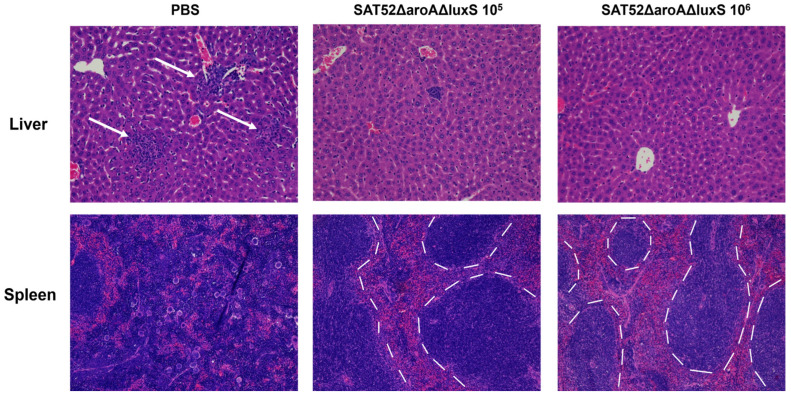
Representative images of histopathological examinations of liver and spleen at 24 h post challenge. Comparing with PBS group, the immunized groups exhibited no obvious lesions in liver and spleen. The white arrows indicate the infiltration of inflammatory cells and formation of inflammatory foci. The dotted lines indicate the boundary between red and white pulps in spleen of immunized mice. The images were taken at 200× magnification.

**Table 1 vaccines-12-00162-t001:** The primers used in this study.

Primers	Sequence (5′ to 3′) ^a^	Usage
aroAMu-F	CACGGCCAGTCTGTGGGGTTTTTATTTCTGTTTTTTGAGAGTTGAGTTTCTGTAGGCTGGAGCTGCTTC	Construction of *aroA* gene mutation
aroAMu-R	TGCGAATACGGACTCGGCGCGCCAGCCCGTCGACTGGCGCAACAGAAGACCATATGAATATCCTCCTTAG	Construction of *aroA* gene mutation
aroAout-F	GAGTTTATCCAGGTCGCTGA	Identification of *aroA* gene mutation
aroAout-R	GCGCATTATACGCGCCAA	Identification of *aroA* gene mutation
aroAin-F	CAATTACACCCTTTCTGCCGA	Identification of *aroA* gene mutation
aroAin-R	GGTCAGGAACTGGCTGGAA	Identification of *aroA* gene mutation
luxSMu-F	AACAAAGAGTTCAGTTTATTTTTAAAAAATTATCGGAGGTGACTAAATGGTGTAGGCTGGAGCTGCTTC	Construction of *luxS* gene mutation
luxSMu-R	CATAAACCGGGGTTAATTTAAATACTGGAACCGCTTACAAATAAGACTACATATGAATATCCTCCTTAG	Construction of *luxS* gene mutation
luxSout-F	AACGTGACGCTTCAGTACGT	Identification of *luxS* gene mutation
luxSout-R	TTAACAGGCCAGGCATTAC	Identification of *luxS* gene mutation
luxSin-F	AGATAGCTTCGCAGTCGATCAT	Identification of *luxS* gene mutation
luxSin-R	TCACTGAGCGAGTGCATCTG	Identification of *luxS* gene mutation
mGAPDHRT-F	GCACAGTCAAGGCCGAGAAT	Real-time PCR for mouse GAPDH
mGAPDHRT-R	GCCTTCTCCATGGTGGTGAA	Real-time PCR for mouse GAPDH
mIFNγ RT-F	TCAAGTGGCATAGATGTGGAAGAA	Real-time PCR for mouse IFNγ
mIFNγ RT-R	TGGCTCTGCAGGATTTTCATG	Real-time PCR for mouse IFNγ
mIL4 RT-F	TCGGCATTTTGAACGAGGTC	Real-time PCR for mouse IL-4
mIL4 RT-R	GAAAAGCCCGAAAGAGTCTC	Real-time PCR for mouse IL-4

^a^ For aroAMu-F/R and luxSMu-F/R pairs, the underlined parts are homologous sequences of target genes, and the rest of the 3′ parts are sequences to amplify antibiotic resistance genes. aroAout-F/R and luxSout-F/R pairs target the flank regions of mutated genes. For aroAin-F/R and luxSin-F/R primer pairs, one oligo primer targets the FRT sequence which replaces the mutated genes, while the other targets the flank region of mutated genes.

**Table 2 vaccines-12-00162-t002:** Determination of LD_50_ for *Salmonella* strains SAT52 and SAT52ΔaroAΔluxS.

Bacterial Strains	Dose of Challenge (CFUs)	No. of Deaths/No. of Total Mice	LD_50_ Value (CFUs)
SAT52	1 × 10^9^	8/8	3.16 × 10^4^
	1 × 10^8^	8/8	
	1 × 10^7^	8/8	
	1 × 10^6^	8/8	
	1 × 10^5^	5/8	
	1 × 10^4^	3/8	
SAT52ΔaroAΔluxS	1 × 10^9^	8/8	4.22 × 10^7^
	1 × 10^8^	7/8	
	1 × 10^7^	0/8	
	1 × 10^6^	0/8	
	1 × 10^5^	0/8	
	1 × 10^4^	0/8	
PBS	/	0/8	/

## Data Availability

The data generated and analyzed during this study are available from the corresponding authors on reasonable request.

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
