# Peer review of "The aroA and luxS Double-Gene Mutant Strain Has Potential to Be a Live Attenuated Vaccine against Salmonella Typhimurium"

_vaccines, 2024, doi:10.3390/vaccines12020162_

Round 1

Reviewer 1 Report

Comments and Suggestions for Authors

The current manuscript “The aroA and luxS double genes mutant strain has potential to 2 be live attenuated vaccine against Salmonella Typhimurium” by Zuo et al. described an attenuated live candidate vaccine targeting Salmonella Typhimurium. The authors showed an analysis of the vaccine strain, immunogenicity, and protection of this vaccine in mice models. There are a few questions that need to be addressed before publication.

1.     The main purpose of this manuscript was to evaluate the safety and protection profiles of the aroA and luxS double mutant because a single aroA mutant had safety concerns in the previous studies. However, there is no direct comparison between the single and double mutants to show the benefit of the additional luxS mutation. In addition, the animal models used in the current manuscript are not the same as those in the previous studies. Thus, there is no way to compare the safety profile for these two mutants. Please add new evidence to explain why double mutant is better.

2.     How was the whole cell lysate prepared? Have the authors measured the level of anti-LPS antibody following immunization?

3.     Statistical analysis in Figure 3 CD was not appropriate. A non-parametric Mann-Whitney t-test should be used, assuming the data is not normally distributed. What did the bar graphs show? Mean±SD? How many samples per group were analyzed? Please specify them in the legend.

4.     Figure 4 has the same suggestion for the statistical analysis as above. Median instead of mean ±SD should be used. 

5.     Please add arrows pointing to the lesions and other pathogenic regions in Figure 5.

Comments on the Quality of English Language

The writing of this manuscript needs to be improved.   Some typos or misconceptions were found in the manuscript, such as:

Line 18, 106 CFU

Lines 44-45, Typhimurium does not have Vi capsule

Line 88, “were preserved in our lab used for genetic deletion” need rewrite

Lines 108-109, “The freshly cultured bacteria were diluted to OD600nm as 1.0 and inoculated into 100 mL LB media, respectively” is hard to understand.

Line 146, why did the authors use goat anti-rat IgG to measure mouse antibody? Where was the antibody purchased from? Is there any cross-activity between rat and mouse?

Line 178, “double mutants were constructed” Should be “double mutant was constructed”

Line 237, “more than 4-fold of PBS group” should be “more than 4-fold than that of PBS group”

Line 288, “plays” should be “play”

Lines 290-291, “But it could not induce strong T-cell immune response, which is required to clean  intracellular bacterial infection”. This is not the only reason Vi polysaccharide vaccine is not working well. Vi polysaccharide can not induce T-cell dependent anti-Vi antibody and will not work in children. There are Vi conjugate vaccines available that can protect both children and adults. 

Line 327, “ which might also enhance mutant strain’ immunogenicity”. Do you mean “which might also enhance mutant strain’s immunogenicity”

Author Response

Responses to reviewer 1's comments

Major comments

  1. The main purpose of this manuscript was to evaluate the safety and protection profiles of the aroA and luxS double mutant because a single aroA mutant had safety concerns in the previous studies. However, there is no direct comparison between the single and double mutants to show the benefit of the additional luxS mutation. In addition, the animal models used in the current manuscript are not the same as those in the previous studies. Thus, there is no way to compare the safety profile for these two mutants. Please add new evidence to explain why double mutant is better.

Response: Thanks for your comment. Actually, we made a comparison of virulence in mouse between aroA single mutant strain and SAT52ΔaroAΔluxS. In our unpublished data, LD50 of aroA single mutant strain was 1.78 × 107 CFU (LD50 of SAT52ΔaroAΔluxS was 4.22 × 107 CFU), and its clearance period in host was much longer, indicating that it has a stronger virulence than SAT52ΔaroAΔluxS.

Besides, according to several published studies, ΔaroA single mutant strains needs much higher vaccination dose to achieve 100% protection rate in mouse model. In David O’Callaghan’s work, LD50 of aroA mutant strain is 107.4 CFU and it provides 100% protection in mice with 108.82 CFU (1). In Peter J Valentine’s work, 1×108 CFU of aroA mutant strain only provided 33% protection rate (2). Thus, comparing with aroA single mutant strain, SAT52ΔaroAΔluxS is safer, more avirulent and provides better protection.

  1. How was the whole cell lysate prepared? Have the authors measured the level of anti-LPS antibody following immunization?

Response: Thanks for your comment. The whole cell lysate of SAT52ΔaroAΔluxS was prepared as follows:

First, SAT52 was cultured in 100 mL of LB media until OD600nm reached around 1.0. Then, the bacterial pellet was collected by centrifuge at 5000 rpm for 10 min. After that, the pellet was resuspend with 10 mL of RPMI-1640 media (Gibco, ThermoScientific, Beijing, China), and lysed with sonication. The lysate was pushed through 0.22 μm sterilized filter, then diluted in cell culture media (RPMI-1640 media with 10% FBS) at 1:10. This part has been included in revised manuscript.

To evaluate humoral immunity induced by vaccination, antibody against whole cell of SAT52ΔaroAΔluxS was measured. Level of antibody against LPS was not specifically measured.

  1. Statistical analysis in Figure 3 CD was not appropriate. A non-parametric Mann-Whitney t-test should be used, assuming the data is not normally distributed. What did the bar graphs show? Mean±SD? How many samples per group were analyzed? Please specify them in the legend.

Response: Thanks for your comment. In all figures, that data is presented as mean ± SD bar. For Fig 3A, 3 mice per group were analyzed. Fog Fig 3B-D, 4 mice per group were analyzed. If we follow your suggestion and use non-parametric Mann-Whitney t-test to analyze the statistical significance between immunization groups and PBS group, the P value is higher than 0.05, despite there are multiple-fold gaps between them. Meanwhile, according to other published studies related to Salmonella live attenuated vaccines, we found that most them use one or two way ANOVA for cytokine mRNA level analysis (3-6). Thus, we suppose the statistical analysis method in this manuscript is appropriate.

  1. Figure 4 has the same suggestion for the statistical analysis as above. Median instead of mean ±SD should be used. 

Response: Thanks for your comment. We redid the statistical analysis for Figure 4B and C with non-parametric Mann-Whitney t-test, both of them yield P values lower than 0.01. And for these two figures, 5 mice per group were analyzed. As mentioned above, according to other published studies, we suppose that values presented as mean ± SD is also appropriate for our study.

  1. Please add arrows pointing to the lesions and other pathogenic regions in Figure 5.

Response: Thanks for your comment. We have added arrows to demonstrate the pathologic lesions in liver caused by challenge. For spleen, since the lesions are featured with unclear boundary between white and red pulp, we just used dashed line in vaccination group to demonstrate the normal boundary in health spleen. The figure legend was also updated. Please refer to Figure 5 for details.

Minor comments

Line 18, 106 CFU

Response: Thanks for your comment. We have corrected and highlighted this part in revised manuscript.

Lines 44-45, Typhimurium does not have Vi capsule

Response: Thanks for your comment. As you mentioned that S. enterica serovar Typhi, instead of S. Typhimurium, has Vi capsule. Thus, Vi polysaccharide is not an ideal choice for S. Typhimurium vaccine development. We have revised and highlighted the corresponding parts in both introduction and discussion sections.

Line 88, “were preserved in our lab used for genetic deletion” need rewrite

Response: Thanks for your comment. We have rewritten and highlighted this part in revised manuscript.

Lines 108-109, “The freshly cultured bacteria were diluted to OD600nm as 1.0 and inoculated into 100 mL LB media, respectively” is hard to understand.

Response: Thanks for your comment. We have rewritten and highlighted this part in revised manuscript.

Line 146, why did the authors use goat anti-rat IgG to measure mouse antibody? Where was the antibody purchased from? Is there any cross-activity between rat and mouse?

Response: Thanks for your comment. We double checked our protocol, it was goat anti-mouse IgG that was used as secondary antibody for ELISA. We have rewritten and highlighted this part in revised manuscript.

Line 178, “double mutants were constructed” Should be “double mutant was constructed”

Response: Thanks for your comment. We have rewritten and highlighted this part in revised manuscript.

Line 237, “more than 4-fold of PBS group” should be “more than 4-fold than that of PBS group”

Response: Thanks for your comment. We have rewritten and highlighted this part in revised manuscript.

Line 288, “plays” should be “play”

Response: Thanks for your comment. We have rewritten and highlighted this part in revised manuscript.

Lines 290-291, “But it could not induce strong T-cell immune response, which is required to clean  intracellular bacterial infection”. This is not the only reason Vi polysaccharide vaccine is not working well. Vi polysaccharide can not induce T-cell dependent anti-Vi antibody and will not work in children. There are Vi conjugate vaccines available that can protect both children and adults. 

Response: Thanks for your comment. Indeed, Vi polysaccharide alone cannot induce effective T-cell dependent anti-Vi antibody. But Vi polysaccharide conjugated vaccine could induce high titer of IgG and IgA, as well as monoclonal antibodies which mediate antibody-dependent monocyte phagocytosis and complement deposition against S. enterica serovar Typhi infection (7). However, since S. Typhimurium has no capsule, the susceptible hosts can hardly benefit from Vi polysaccharide vaccination. Thus, we have rewritten the corresponding part in discussion section in revised manuscript.

Line 327, “ which might also enhance mutant strain’ immunogenicity”. Do you mean “which might also enhance mutant strain’s immunogenicity”

Response: Thanks for your comment. We are truly sorry for this typo in original manuscript, we have corrected this in revised manuscript.

We appreciate your effort in improving this manuscript!

References

  1. O'Callaghan D, Maskell D, Liew FY, Easmon CS, Dougan G. Characterization of aromatic- and purine-dependent Salmonella typhimurium: attention, persistence, and ability to induce protective immunity in BALB/c mice. Infect Immun. 1988;56(2):419-23.
  2. Valentine PJ, Devore BP, Heffron F. Identification of three highly attenuated Salmonella typhimurium mutants that are more immunogenic and protective in mice than a prototypical aroA mutant. Infect Immun. 1998;66(7):3378-83.
  3. Kang X, Yang Y, Meng C, Wang X, Liu B, Geng S, et al. Safety and protective efficacy of Salmonella Pullorum spiC and rfaH deletion rough mutant as a live attenuated DIVA vaccine candidate. Poult Sci. 2022;101(3):101655.
  4. Park S, Jung B, Kim E, Hong ST, Yoon H, Hahn TW. Typhimurium Lacking YjeK as a Candidate Live Attenuated Vaccine Against Invasive. Front Immunol. 2020;11:1277.
  5. Senevirathne A, Hewawaduge C, Lee JH. Attenuated Salmonella secreting Brucella protective antigens confer dual-faceted protection against brucellosis and salmonellosis in a mouse model. Vet Immunol Immunopathol. 2019;209:31-6.
  6. Lloren KKS, Lee JH. Live-Attenuated Salmonella-Based Oral Vaccine Candidates Expressing PCV2d Cap and Rep by Novel Expression Plasmids as a Vaccination Strategy for Mucosal and Systemic Immune Responses against PCV2d. Vaccines (Basel). 2023;11(12).
  7. Dahora LC, Verheul MK, Williams KL, Jin C, Stockdale L, Cavet G, et al. Typhi Vi capsule prime-boost vaccination induces convergent and functional antibody responses. Sci Immunol. 2021;6(64):eabj1181.

Reviewer 2 Report

Comments and Suggestions for Authors

 This manuscript describes the potential use of a strain of Salmonella with mutation in aroA and luxS genes as a live attenuated vaccine against Salmonella Typhimurium. 

Merit

1. The development of an effective vaccine against Salmonella Typhimurium is essential as it would lead to the control of the incidence of Salmonellosis in animals and humans. The authors of this manuscript have provided preliminary evidence suggesting that the double mutant strain of Salmonella Typhimurium may be a good candidate for vaccine.    

Comments

1.      Lines 45 – 66: the authors did a good job presenting efforts of other researchers in addressing Salmonella vaccine. The authors should clearly state why their approach may be better than the similar studies reported to date.

2.      The reasons for targeting the aroA and luxS genes should be stated in the introduction section. 

3.      There are several sentence/grammar issues that should be addressed (Lines 87 – 88 as an example). 

4.      Section 2.4; lines 106 – 110: OD monitoring without determination of the number of colony forming units (CFUs) of the wild type and mutant strain is not sufficient. Confirmation of colony forming unit in the starting material is necessary to at least assure that equal number of both strains are used. This will reduce bias.    

5.      Section 2.6, line 129: specify the exact number of CFUs of bacteria used and why it is necessary to use different CFUs.   

6.      Section 2.10, line 163: How many mice per group? 

7.      The figures and images are clear, good quality, and easy to follow. 

8.      Methods are standard and I have no issues with them.

Comments on the Quality of English Language

Several sentences need improvement. 

Author Response

Responses to reviewer 2

Major comments

  1. Lines 45 – 66: the authors did a good job presenting efforts of other researchers in addressing Salmonella vaccine. The authors should clearly state why their approach may be better than the similar studies reported to date.

Response: Thanks for your comment. I would like to emphasis the advantage of our development strategy in two ways. First, to knockout the desired genes, λ-Red homologous recombination was used. It is an effective and specific method to achieve targeted chromosomal insertions or deletions in Escherichia and Salmonella without residual antibiotic cassette. And our data shows that the transcription of flank genes are not affected, indicating the specificity and reliability of this gene editing method. Second, double genes mutation was used to further attenuate live vaccine’s virulence, while maintain its immunogenicity as much as possible. However, currently, few commercial vaccines with double genes mutation are available. In this way, our study offers a potential candidate for future vaccine development. Besides, comparing with other double genes mutant strains in published studies (1), our SAT52ΔaroAΔluxS exhibits even lower pathogenicity in hosts, and offers good protection rate with lower vaccination dose, making it a promising choice. This has been added in discussion section of revised manuscript.

  1. The reasons for targeting the aroA and luxS genes should be stated in the introduction section. 

Response: Thanks for your comment. In our previous study, it was that deletion of both aroA and luxS would further attenuate Escherichia coli virulence in susceptible hosts, without significant impact on its immunogenicity. It intrigued us to explore whether double deletion of aroA and luxS would produce a potential live attenuated vaccine candidate in Salmonella. We have added this part in revised manuscript.

  1. There are several sentence/grammar issues that should be addressed (Lines 87 – 88 as an example). 

Response: Thanks for your comment. We have went through this part to correct any sentence/grammar issues in revised manuscript.

  1. Section 2.4; lines 106 – 110: OD monitoring without determination of the number of colony forming units (CFUs) of the wild type and mutant strain is not sufficient. Confirmation of colony forming unit in the starting material is necessary to at least assure that equal number of both strains are used. This will reduce bias.    

Response: Thanks for your comment. According to our record, for freshly cultured SAT52, the CFU in 1 mL was 4.96×108 when OD600 reached 1.0. For SAT52ΔaroAΔluxS, the number was 5.02×108. Thus, it’s basically the same starting amount for these two strains in growth assay.

  1. Section 2.6, line 129: specify the exact number of CFUs of bacteria used and why it is necessary to use different CFUs.   

Response: Thanks for your comment. According to Spearman-Karber method, to determine the LD50 of SAT52 and SAT52ΔaroAΔluxS, multiple groups of mice must be infected with different doses of bacteria, respectively. The exact CFU numbers for each group are listed in Table 2. More details in section 2.6 are also included in revised manuscript.

  1. Section 2.10, line 163: How many mice per group? 

Response: Thanks for your comment. Sixteen mice are included in each group. More details in section 2.10 are included in revised manuscript.

  1. The figures and images are clear, good quality, and easy to follow. 

Response: Thanks for your comment.

  1. Methods are standard and I have no issues with them.

Response: Thanks for your comment.

We appreciate your effort in improving this manuscript!

References

  1. Mercado-Lubo R, Gauger EJ, Leatham MP, Conway T, Cohen PS. A Salmonella enterica serovar typhimurium succinate dehydrogenase/fumarate reductase double mutant is avirulent and immunogenic in BALB/c mice. Infect Immun. 2008;76(3):1128-34.

Round 2

Reviewer 1 Report

Comments and Suggestions for Authors

The authors have successfully answered all my questions.